# Effect of Vibration Massage and Passive Rest on Recovery of Muscle Strength after Short-Term Exercise

**DOI:** 10.3390/ijerph182111680

**Published:** 2021-11-07

**Authors:** Wiesław Chwała, Paweł Pogwizd, Łukasz Rydzik, Tadeusz Ambroży

**Affiliations:** 1The Department of Biomechanics, University of Physical Education, Al. Jana Pawla II 78, 31-571 Cracow, Poland; wieslaw.chwala@awf.krakow.pl; 2Research and Development Department of Vitberg, Marcina Borelowskiego 29, 33-300 Nowy Sącz, Poland; ppogwizd@o2.pl; 3Institute of Sports Sciences, University of Physical Education, 31-571 Krakow, Poland; tadek@ambrozy.pl

**Keywords:** recovery, fatigue, muscle, vibration, massage, sport, training

## Abstract

Background: The aim of the study was to compare the effect of vibration massage and passive rest on accelerating the process of muscle recovery after short-term intense exercise. Methods: Eighty-four healthy men aged 20 to 25 years participated in the study. Study participants performed isometric (ISO-M Group) and auxotonic (AUX-M group) contraction exercise in the lower limbs. Vibration massage was administered after exercise in the first recovery period. In the same period, controls rested passively, without the support of vibration massage. To assess the effectiveness of the applied vibration, a 4-fold measurement of the maximum force of the muscles involved in the exercise was performed under conditions of isometric contractions on a leg press machine set at an angle of 45° degrees upwards. Results: Differences in maximum strength during isometric contraction were found compared to baseline in favor of the groups subjected to the experimental vibration massage. Differences were demonstrated in muscle strength between the study groups (*p* < 0.005). The second period of passive rest in all groups did not bring significant changes in the values of maximal lower limb strength. Conclusions: Properly selected characteristics of the vibration effect can be an effective method in accelerating recovery and regaining lost motor capabilities of muscle groups fatigued by exercise. This offers the potential to shorten rest periods between sets of repetitions in training or between training units.

## 1. Introduction

Intense physical activity leads to fatigue, resulting in a deterioration of exercise capacity, manifested by a decrease in intensity and efficiency of muscle work [1,2]. In the case of maximal physical exercise, muscle fatigue is experienced from the first seconds [3,4]. The magnitude and nature of the functional changes in the body induced by physical exercise are largely influenced by the intensity, duration, and specificity of the exercise [5]. They are also affected by the size and type of muscle groups involved in performing the specific work, the type of contraction of the muscles involved, and the individual body metabolism [6].

Researchers point to possible differences during isometric and auxotonic muscle work. During auxotonic and isometric exercise, alternating muscle contraction and relaxation facilitate blood circulation, which supplies the intensely working muscles with oxygen and energy substrates and helps remove metabolic products from them [7,8,9].

The situation is different when muscles work under isometric conditions. During isometric exercise, muscles put pressure on blood vessels. Isometric muscle work increases the rate of fatigue, which means a reduction in the ability to perform exercise [4]. As a result, even at low loads of 25–30% of the maximum voluntary contraction (MVC), venous blood outflow from the muscles is limited [10]. This is despite a significant increase in blood pressure [7,8]. The free flow of blood is restricted, which in turn interferes with the supply of essential nutrients and the removal of metabolic products.

Athletes strive to reduce the effects of fatigue and delay the onset of its symptoms. Fatigue increases during each training unit, after successive sets of exercises, especially during exercises at maximal or submaximal intensity [11]. This is due to depletion of energy stores and significant acidification of the body [12]. The state of increasing fatigue determines the time of recovery breaks between successive sets and the frequency of training units of a specific exercise. The use of effective methods of athletic recovery can significantly accelerate the process of regaining body performance and increase the effectiveness of training. Various methods and techniques are used to improve the effectiveness of post-exercise recovery. One criterion for evaluating the effectiveness of the recovery method used is whether a faster recovery to the baseline levels can be observed compared to that after passive rest [13]. Today, a number of methods are used to accelerate recovery, including different techniques of massage, which is a therapeutic procedure used for a long time in sports [14]. Research shows that using massage before competitions improves athletic performance [15]. The use of massage after exercise reduces muscle soreness and helps prevent potential injuries [16]. It has also been demonstrated that massage significantly modifies the length of tendons [17]. In sports, massage is one of the most popular forms used to accelerate athletic recovery. The effect of mechanical whole-body vibration (WBV) on the body, which was used in the present study, is similar to classical massage using vibration techniques.

It is recognized that an increase in blood flow resulting from the application of WBV can accelerate the process of post-exercise recovery by enhancing nutrient exchange [18,19], removing metabolic by-products that inhibit tissue repair, and improving the efficiency of oxygen supply between capillaries and the fluid surrounding the body cells [5,20]. Furthermore, the increase in blood flow produces a thermal effect [21,22], initiating a healing response in tissues damaged during exercise, which can be further enhanced by the heat produced by vibrating muscle fibers [23,24,25,26].

Previous research results have indicated discrepancies between individual authors as to the effects of vibration on restoring strength and speed capabilities of muscles in the recovery period following intense exercise. Studies [27,28,29,30] have shown that vibration is effective in relieving exercise-induced muscle pain. Unfortunately, these studies provide limited information on how the reduction of exercise-induced muscle pain affects the rate at which muscle strength returns to baseline or hypercompensation of muscle strength.

A study by Barnes et al. [31] failed to confirm the positive effects of vibration applied after eccentric contractions of the lower limb muscles on an isokinetic dynamometer. Dabbs et al. [32] also argued that vibration does not help with muscle recovery, which does not translate into improvements in the level of variables that characterize vertical jumps on the platform. In contrast, Annino et al. indicated that the use of WBV during rests between sets delays muscle fatigue and results in better competitive performance of athletes [33].

This study aimed to compare the two forms of recovery (with vibration and passive rest) in supporting muscle recovery after a single submaximal exercise (isometric and auxotonic).

## 2. Materials and Methods

### 2.1. Participants

The study was a randomized control trial with repeated measures. The study involved 84 young men aged 20 to 24 years who had not suffered from orthopedic injuries within a year of the study and had not suffered from other conditions that could significantly affect the results of maximum muscle strength measurements. The participants who were enrolled in the study participated in weekly recreational physical activity of similar intensity, volume, and nature of sporting activities. The characterization of study participants is presented in Table 1. Prior to participation in the tests, the competitors were informed about the research procedures, which were in accordance with the ethical principles of the Declaration of Helsinki WMADH (2000). Obtaining the competitors’ written consent was the condition for their participation in the project. The research was approved by the Bioethics Committee (No. (KB/245/FI/2020).

### 2.2. Procedure

Eighty-four participants were randomized into 4 equal groups. Each respondent drew lots to determine their group. Each study group was tested by the same researchers, using the same measurement tools and identical measurement procedures. The experimental design is shown in Figure 1.

Each participant began the activity with a standard warm up consisting of 15 alternating repetitions of flexion and extension of the lower limb joints on a leg press machine (Body Craft F660, Fort Lauderdale, OH, USA) in a seated position (the back at 45 degrees to the ground), with the machine set at an angle of 45° degrees upwards. The load was selected individually and was 30% of one repetition maximum (1RM); it was tested one day prior to the actual testing. The whole procedure was repeated three times. In the starting position, the angle of limb flexion at the knee joints measured between the thigh and the shank was monitored with a goniometer and was 80° (0°: full joint extension). The ankle joints in the initial setting were in a neutral position (90° angle between the foot and the lower leg). The angle at the hip joint was determined by the flexion angles of the knee and ankle joints and the individual dimensions of the lower limb segments. The adjustment of the leg press machine ensured constant angle settings in the knee and ankle joints. The pelvis and upper body were stabilized with straps.

To determine the baseline maximum level of strength of the lower limb muscles, a measurement of the maximum force of isometric contraction was carried out after the warm up on a measuring stand in a seated position, with the leg press machine set at an angle of 45° degrees upwards and the lower limb joints positioned in the starting position, as described above.

In order to determine the maximum level of strength of the lower limb muscles, after the warm up using leg press machine (Body Craft F660, Fort Lauderdale, OH, USA), repetitions with maximum force were performed. The crane was set at 45° and the working procedure was as described above. 

Next, each subject was subjected to submaximal physical exercise that loaded the lower limb muscle groups, in particular, aimed at their fatigue. Groups ISO-M and ISO-P performed exercise consisting of 3 min isometric exercise following the warm up of the lower limb muscles during one 60-s submaximal contraction at 75% of individual maximal strength, in a seated position with the leg press machine set at an angle of 45° degrees upward.

Groups AUX-M and AUX-P performed one auxotonic exercise, consisting of 20 repetitions at the level of 75% of individual maximum strength without recovery breaks by alternate extension and flexion of the joints of the lower limbs with the upper body and pelvis stabilized by straps, on a leg press machine in a seated position, with the machine positioned at 45° degrees upward. The flexion angles of the lower limbs in the starting position were identical to those during the measurement of maximal strength.

Immediately after the completion of the exercise, the maximum strength of the muscles involved in the exercise was measured a second time in the experimental and control groups.

Next, participants from experimental groups ISO-M and AUX-M were subjected to the first 20-min recovery period with the use of vibration sessions. Study participants from control groups ISO-P and AUX-P recovered passively without vibration sessions in the same position as study participants from groups ISO-M and AUX-M.

The third measurement of the level of maximal strength of the lower limb muscles was performed after the first recovery period.

After the measurement, the second 20 min of post-exercise recovery was started (passive rest in all groups).

The final (fourth) measurement was taken immediately after the end of the second 20-min passive rest in all groups.

### 2.3. Vibration Sessions

Vibration massage of the involved muscles was performed during rest after submaximal exercise in groups ISO-M and AUX-M. The vibration was generated by a device designed at Vitberg. An author’s vibration program was developed specifically for the study, which, during a 20-min procedure, generated vibrations in the frequency range of 20 to 50 Hz, amplitude < 0.5 mm, and varied in terms of the duration of intervals (from 1 to 4 s) [21,34]. The characteristics of the vibration intervention also took into account the fact that there was a possibility of individual variation in adaptation to the applied vibration frequency, resulting from individual characteristics and properties of the neuromuscular spindles, different numbers and locations of mechanoreceptors and proprioreceptors, or different elastic characteristics of the muscle-tendon complex and the proportion of type II fibres in the muscle [35]. The vibrating mattress and vibrating module were used in the study, with their total application surface allowing for simultaneous (without changing the position of the mattress and the participant during the procedure) massaging of the muscle groups of both lower limbs engaged previously in the exercise (Figure 2).

### 2.4. Statistical Analysis

Statistical analysis was performed using the STATISTICA 13.3 PL software. The exercise-induced changes in maximal leg strength during both periods of recovery were used to determine their absolute increase compared to baseline (measured after the warm up) in subsequent periods of the experiment. The significance of statistical differences between the maximum relative values of changes in muscle strength was tested by ANOVA for repeated measures and using Tukey’s post-hoc test. Friedman ANOVA test with Dunn’s post-hoc test was used in the case of the lack of normal distribution. The significance of differences between maximum relative values of increments of the variables analyzed in all groups was assessed by one-way analysis of variance ANOVA using Tukey’s post-hoc test. In the case of the lack of normal distribution or lack of homogeneity of variance tested by Levene’s test, ANOVA Kruksal–Wallis test was applied, followed by Dunn’s post-hoc test. For variables with values showing the characteristics of a normal distribution, the arithmetic mean and standard deviation (± s) are given, while for variables whose values did not have a normal distribution, in addition to the means, the median and quarter deviation (Me ± Q) are also reported. Furthermore, a measure of the effect size was calculated by presenting eta-squared values.

## 3. Results

Table 2 shows the basic statistical variables (ΔFwmax_w-p, ΔFwmax_r1-p, ΔFwmax_r2-p), expressed in Nkg^−1^ for all groups studied.

Figure 3 shows the percentage changes of the maximum relative strength of the examined muscles between successive periods for each group compared to the baseline taken as 100%.

The mean values of changes in ΔFwmax recorded after each period of the experiment relative to baseline were then subjected to the analysis of the statistical significance of differences for repeated measures in all groups of participants. Strong effect size was also demonstrated in most measurements. The results of the post-hoc tests and effect size are presented in Table 3.

The results of the analysis of variance in intergroup comparisons indicate the presence of statistically significant differences in the variable studied only within some pairs of groups of participants in subsequent periods of the experiment. Between-group comparisons revealed strong measures of effect size. As a result of post-hoc testing and effect size, groups of participants with the statistically significantly different values of the variable analyzed were separated as presented in Table 4.

The post-exercise changes in maximum relative strength ΔFwmax_w-p between the different groups of participants are shown in Figure 4, Figure 5 and Figure 6.

## 4. Discussion

In the ISO-M group of participants, who performed submaximal isometric exercise and underwent vibration massage during the recovery period, a statistically significant (*p* < 0.001) and most substantial increase in the maximum relative strength of about 4.8 Nkg^−1^ was observed after the first recovery period, which corresponded to an average increase of about 16% compared to the value of post-exercise maximum relative strength (Table 2 and Figure 3). It can therefore be concluded that the increase in strength is related to the application of vibration massage. Fuller et al. claim that vibration massage has similar efficacy to classical massage [36]. Furthermore, in the AUX-M group, subjected to auxotonic exercise and experimental vibration exposure, an average increase in strength of approximately 12% was recorded during an identical period of the experiment, which accounted for an increase in the relative strength of 3.6 Nkg^−1^ compared to the value of the variable recorded after the exercise test (Table 2 and Figure 3). In addition, in this case, the results of the statistical analysis confirmed the presence of significant differences at *p* < 0.001. This may indicate the efficacy of the vibration massage in muscle recovery resulting in increased strength. Similar findings were reported by Mukhtar et.al, who showed that vibration therapy had a significant positive effect on neuromuscular performance, resulting in improved upper extremity strength [37].

It should be added that in the groups subjected to vibration massage, the values of maximum relative strength in maximum isometric contraction exceeded the baseline levels already after the first period of recovery. The recovery of previously lost relative strength, together with the occurrence of the hypercompensation effect, was noted in both groups (by 3.5% in the ISO-M group and 1.2% in the AUX-M group), even though they differed in the form of physical exercise to which the muscles were subjected. Scientific research has shown that vibration therapy is an effective recovery method [38].

In the ISO-P group (isometric contraction/passive rest), there was also a statistically significant (*p* < 0.01) increase in relative strength after the first recovery period. However, it was not as significant as in the groups with vibration massage and averaged to about 4.4%, which translated into an increase in the maximal relative force of 1.5 Nkg^−1^ after the first recovery period (Table 2 and Figure 3). In conclusion, the process of restoring relative strength after an analogous isometric exercise in the first recovery period in the group with vibration massage was more than three times faster than in the group with passive rest.

Furthermore, in participants from the AUX-P group (auxotonic muscle contraction/passive rest), despite a 7% increase in relative strength observed after the first recovery period (Figure 3) compared to the value recorded after the exercise test, the statistical analysis did not confirm the presence of statistically significant differences. Similarly as in the case of submaximal isometric contractions, the effectiveness of the vibration massage was confirmed in the statistical comparisons determining the rate of restoring the lost relative strength following auxotonic exercise. The efficiency of recovery, expressed by the restoration of the relative strength, was similarly about three times higher than in the case of passive rest. Our study confirmed the differences in the increase in relative strength after the first recovery period between ISO-M, ISO-P, AUX-M, and AUX-P groups. The recorded increases in maximum strength induced by vibration massage can be explained by muscle recovery that occurs as a result of muscle vibration [38,39].

The different observations of the authors of studies [31,32,36,40,41,42,43] regarding the evaluation of the effect of vibration massage aimed to ensure athletic recovery of muscles that have been previously fatigued during high-intensity exercise may be due to several factors. The nature and pattern of the vibration massage may be an important problem. This problem has also been noticed by Cochrane and Booker and Park who believe that following vibration, any changes related to muscle motor activity depend on the mutual relations between vibration parameters such as frequency (Hz), amplitude (mm), and vibration duration (s) [44,45]. A direct comparison of these studies is difficult due to the lack of homogeneity of the physical effort, the varied measurement methodology, the characteristics of the vibrations produced, and the variety of positions in which the vibration massage was performed. All this may explain the considerable discrepancies in the results obtained by individual researchers in the studies cited.

Previous results reported by various authors indicate a considerable scatter concerning the optimal parameters of vibration stimulus suggested by them. Consequently, there are no unequivocal guidelines for the effective choice and application of vibration parameters. Issurin suggests that low-frequency vibration massage (f = 15–50 Hz) increases the local temperature of the tissues subjected to vibration, causing relaxation of myofascial tissues, a decrease in emotional tension, and a general calming effect [21]. On the other hand, a massage using high-frequency vibrations leads to an increase in excitability of the central nervous system, increases muscle tension, and has a rapid warming effect. However, according to the same author, a massage with the use of both low- and high-frequency vibrations brings positive effects. Despite this, Issurin [21] recommends that the massage time for high-frequency vibration should be much shorter than in the case of lower-frequency vibration stimulus. In our study, the vibration time also decreased with increasing vibration frequency. Furthermore, Rittweger et al. reported that vibration frequencies below 20 Hz lead to relaxation in muscles, while frequencies over 50 Hz can cause acute muscle pain in untrained individuals [46]. There are also reports which indicate that the vibration frequency range of 30–50 Hz is most effective to activate muscle fibers [47]. Ronnestad suggested that the optimal vibration frequency for professional athletes should be 50 Hz [48]. Low-frequency vibrations of 5–15 Hz, on the other hand, can speed up the process of post-exercise recovery due to increased blood flow to and from the injured muscle and stimulate muscle receptors to relieve muscle tension [34].

A very interesting suggestion was made by Lamont et al. who recommend starting vibration sessions with lower frequencies and amplitudes and gradually increasing them while shortening the exposure to vibrations [49]. The vibration interaction algorithm in our study also started from the lowest frequencies. However, the vibration based on 26 and 30 Hz frequencies used in the studies by Bullock et al. [50], Barnes et al. [31], and Dabbs et al. [32] did not produce the expected beneficial effects. Furthermore, vibration with frequencies of 20, 35, and 50 Hz, used in the studies by Marin et al., produced positive results during post-exercise recovery [23]. Higher vibration frequencies suggested by Issurin [21] did not allow Lau and Nosak [27] and Fuller et al. [36] to demonstrate the effectiveness of the vibration procedure during the recovery period after exercise. The presented facts may suggest that vibration frequency is not the only parameter that significantly affects the outcomes of vibration sessions performed during post-exercise recovery. The results of our study provide compelling evidence that different frequencies and amplitudes of vibrations may be required, depending on the expected effects of vibration sessions.

The study used a vibration stimulus with completely different frequency characteristics than those used in previous experiments. No publications to date have presented studies in which the frequency and amplitude of a vibration stimulus are varied smoothly during a single vibration program. During the determination of the characteristics of the vibration intervention, it is critical to remember that the time of recovery of motor skills lost due to exercise depends on many factors: age, muscle mass, type of muscle fibers, level of previous fatigue, fitness level, and specific personal susceptibility to vibration interventions [51]. The mechanism developed by the authors for the effect of the vibration stimulus was partially similar to the suggestions made by Lamont et al. [49]. Another key factor in increasing the effectiveness of vibration massage used in our study was the variation of rest duration (1–4 s) in generating the vibration stimulus in successive sets of vibration in the control program. This view was also noted by Issurin [21], who pointed to the positive effect of vibration stimuli only when intermittent vibration massage was applied. However, subjecting participants to continuous and prolonged vibration massage led to a noticeable decrease in the values of biomechanical variables in the muscles tested.

The characteristics of low-amplitude vibration stimuli proposed in our study were intended to induce a relaxing effect rather than reinforce a stress factor for pre-fatigued muscles. This need was also emphasized by Cardinale and Lim [35]. Furthermore, the support of muscle recovery was supposed to be based on increased blood flow as mentioned by Weerapong et al. [16]. A study conducted by Kerschan-Schindl et al. [52] found an increase in mean blood flow rate in the femoropopliteal artery as a result of vibration massage performed on a vibration platform (f-26 Hz A-3 mm). An increase in local blood flow rate immediately after the vibratory stimulus was also confirmed by other researchers [21,53]. The explanation for this mechanism is based on the assumption that rhythmic muscle contractions are observed when the body is subjected to vibration [54], which can cause changes in the peripheral arteries. According to Kerschan-Schindl et al. [52], the increase in blood flow rate may be due to a decrease in blood viscosity and vasodilation induced by vibration. Therefore, the increase in blood flow rate resulting from the applied vibration can accelerate the process of post-exercise recovery by enhancing nutrient exchange, removing metabolic by-products [40] that inhibit tissue repair and improving the oxygen supply between capillaries and the fluid surrounding body cells [55]. The low value of vibration amplitude was further intended to eliminate the possible adverse phenomenon of potentiation of muscle fatigue due to the vibration procedure, as mentioned by Barnes et al. [31]. The results of the study conducted by Zoladz et al. [56] show that already after a 10-s maximal power exercise test on an isokinetic cycle ergometer, a 25% loss of maximal power was observed, indicating the onset of pronounced fatigue. During short-term dynamic efforts, the immediate energy supply to muscles is obtained from adenosine triphosphate (ATP) and phosphocreatine (PCr) [57,58]. Unfortunately, their content in muscle cells is limited. During the first seconds of exercise, the content of ATP and phosphocreatine in muscle cells decreases rapidly. In the study by Hultman and Sjoholm [59], stimulation of the muscle to contraction for 2.5 s resulted in 26% depletion of PCr stores. Phosphocreatine resynthesis and elimination of glycolytic products occur partly during and partly following the exercise [60,61,62]. Therefore, symptoms of fatigue are observed even during short-term efforts of submaximal intensity [63,64,65]. Another key factor in increasing the effectiveness of vibration massage used in our study was the use of breaks in generation of the vibratory stimulus during the program. This was also observed by Issurin [21], who indicated a positive effect of vibration only when intermittent vibration sessions were applied.

In our study key information was provided by the recorded results of changes in the maximum relative strength of the tested muscle groups, which were recorded at the end of the first and second recovery periods. Based on these results, it was found that vibration sessions conducted in the first period of recovery significantly accelerated the process of post-exercise recovery and restoration of strength capabilities of the examined muscles. In the case of groups where recovery during each period consisted solely of passive rest, even a time of 40 min was insufficient to recover the previous strength capabilities lost due to exercise.

The use of vibration massage resulted in much more effective muscle recovery in the case of fatigue caused by isometric exercise. This is confirmed by the results of the recorded changes in strength after each recovery period, with their rate of return to pre-exercise levels and the possible effect of hypercompensation being significantly higher for the groups subjected to isometric exercise. The reason for this may be the fact of lower regression of energy substrate levels recorded during isometric exercise as demonstrated by Konturek [66] and Ortega et al. [67]. During static exercise, tense muscles put pressure on blood vessels, thus impeding the blood flow, which in turn interferes with the supply of essential nutrients and the removal of metabolic products [7,8]. Consequently, during static muscle work, conditions are created for oxygen debt and an increase in the proportion of anaerobic metabolism causing their acidification [68,69]. Furthermore, the increase in blood flow due to the application of vibration intervention [21,53] facilitates the exchange of nutrients and the removal of metabolic by-products [40,55] which were accumulated during the exercise with isometric muscle work. According to Lattier et al. [70], a very important factor in the acceleration of post-workout relaxation and ability to undertake the next effort is lactate removal from blood and muscles. This view is also supported by other researchers [71,72], who believe that blood lactate accumulation is at least partly responsible for the delayed return to homeostasis after fatigue. Therefore, this helps explain why the vibration interaction in our study led to better recovery after isometric exercise.

Post-exercise rest is an essential part of ensuring body homeostasis. There are scientific studies which stress that in the case of fatigue caused by physical exertion, the athlete should rest actively. Research [73,74] has shown that light exercises, applied after physical exercise, can accelerate the recovery from muscle fatigue and is more effective than passive rest. In a study [75], a higher rate of lactate decomposition was observed during active recovery compared to passive rest. The authors explain this fact by the increased blood flow that occurs during active rest, which in turn facilitates the removal of previously accumulated lactate. Ahmaidi et al. [76] and Thiriet et al. [77] also found that low-intensity active rest between repeated sets of intense exercise results in lower blood lactate concentration compared to passive rest. According to Bush et al. [78] whole body vibration (WBV) induces small muscle contractions during which there is a change in muscle length. Barnes et al. [31] believe that, consequently, muscles subjected to vibration massage are forced to do extra work, which can be a form of active rest after exercise.

## 5. Conclusions

Based on the results obtained in the present study, it seems right to state that the use of our vibration procedure, aimed at previously fatigued muscle groups, is an effective method to accelerate the process of their recovery and regaining the lost motor abilities. Properly chosen parameters of the vibration procedure during active inter-exercise muscle recovery can significantly shorten the necessary breaks in training units and between training sessions and improve training efficiency.

## Figures and Tables

**Figure 1 ijerph-18-11680-f001:**
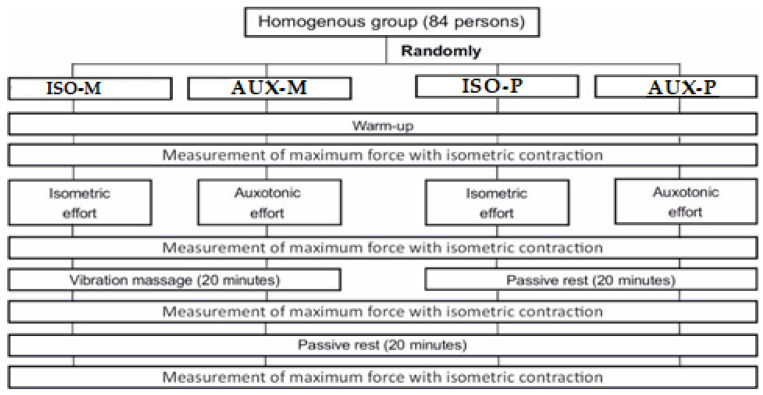
Experimental design.

**Figure 2 ijerph-18-11680-f002:**
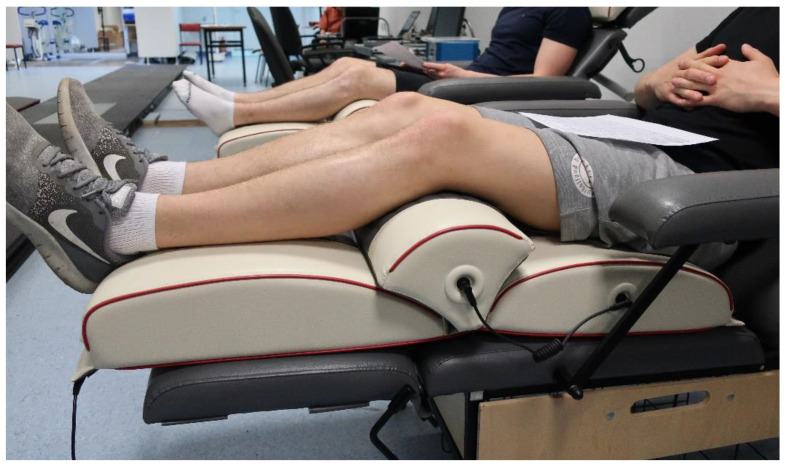
Position of the test subject during vibration massage application. Source: Figure taken by the authors of the paper.

**Figure 3 ijerph-18-11680-f003:**
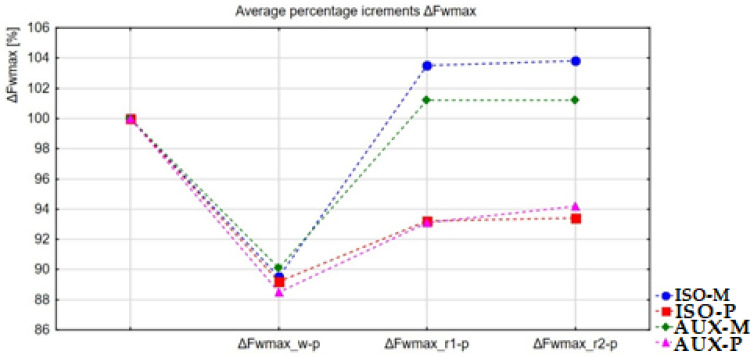
Graphical representation of changes in the maximum relative strength compared to baseline expressed in % in individual periods of the experiment in groups of participants. ΔFwmax_w-p—post-exercise change in relative strength following exercise relative to baseline, ΔFwmax_r1-p—change in relative strength after the first recovery period, relative to baseline, ΔFwmax_r2-p—change in relative strength after the second recovery period, relative to baseline.

**Figure 4 ijerph-18-11680-f004:**
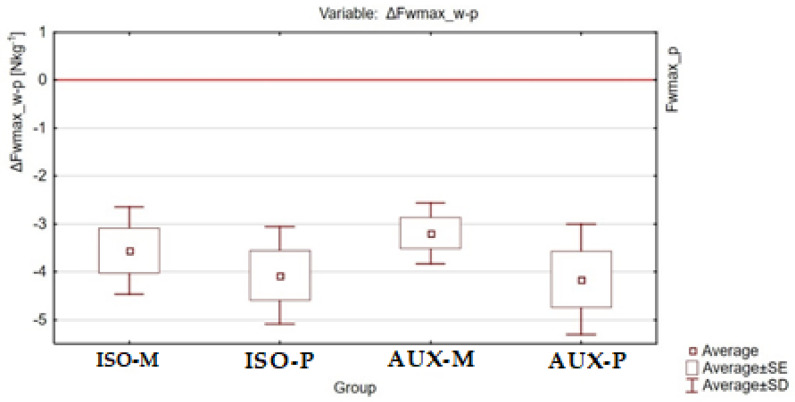
Graphical representation of the results of ANOVA Kruskal–Willis analysis for the ΔFwmax_w-p variable between all tested groups of participants.

**Figure 5 ijerph-18-11680-f005:**
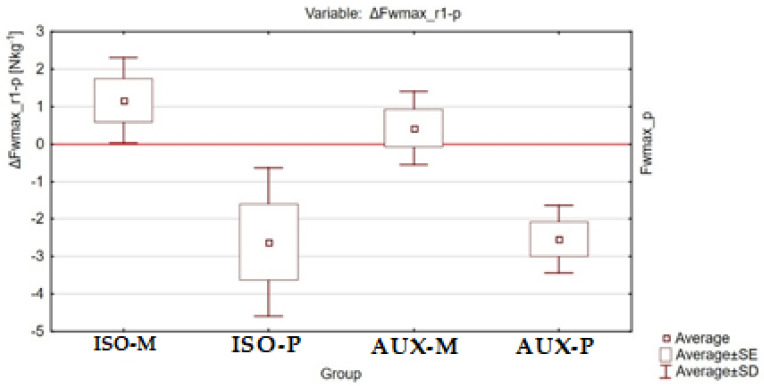
Graphical representation of the results of ANOVA Kruskal–Willis analysis for the ΔFwmax_r-p variable between all tested groups of participants.

**Figure 6 ijerph-18-11680-f006:**
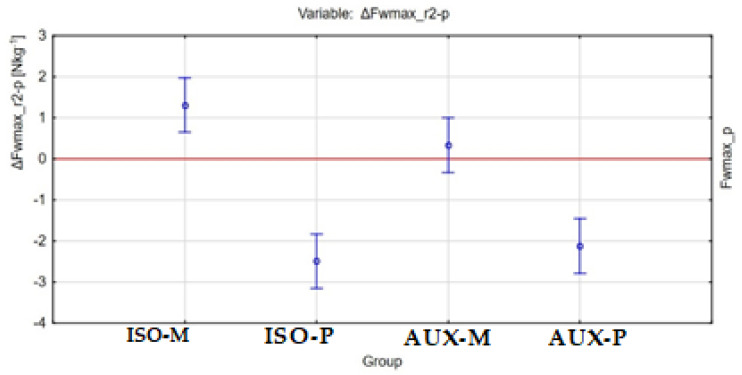
Graphical representation of the results of ANOVA (F-test) for the ΔFwmax_r2-p variable between all tested groups of participants.

**Table 1 ijerph-18-11680-t001:** Characteristics of study participants by group.

Group/Variable	ISO-M	ISO-P	AUX-M	AUX-P
Number [n]	21	21	21	21
Age [years]	20.4 ± 1.78	19.9 ± 1.91	21.1 ± 1.24	20.7 ± 1.55
Body mass [kg]	79.1 ± 11.33	74.9 ± 5.67	72.4 ± 8.34	78.6 ± 8.31
Body height [m]	1.81 ± 0.97	1.79 ± 0.81	1.78 ± 0.86	1.81 ± 0.10
BMI [kgm^−2^]	24.1 ± 1.26	23.2 ± 1.96	22.9 ± 2.04	23.9 ± 1.13

**ISO-M**—experimental group subjected to the isometric muscle exercise using vibration sessions in the first period of recovery and passive rest in the second, **AUX-M**—experimental group subjected to the auxotonic muscle exercise using vibration sessions in the first period of recovery and passive rest in the second, **ISO-P**—control group subjected to isometric exercise using only passive recovery in both periods intended for recovery, **AUX-P**—control group subjected to auxotonic exercise using only passive recovery during the periods intended for recovery.

**Table 2 ijerph-18-11680-t002:** Changes in the maximum relative strength compared to baseline expressed in Nkg^−1^, in individual stages of the experiment in groups of participants.

Variables/Parameters/Group		ISO-M	ISO-P	AUX-M	AUX-P
ΔFwmax_w-p [Nkg^−1^]	$\bar{x}\;$± s	−3.6 ± 2.13	−4.1 ± 2.38	−3.2 ± 1.49	−4.2 ± 2.69
	*Me ± Q*	-	−4.2 ± 1.17	-	−2.9 ± 2.22
	Min.	−8.1	−11.4	−5.9	−8.7
	Max.	−0.4	−1	−0.6	−0.6
ΔFwmax_r1-p [Nkg^−1^]	$\bar{x}\;$± s	1.2 ± 2.65	−2.6 ± 4.64	0.4 ± 2.27	−2.5 ± 2.10
	*Me ± Q*	1.3 ± 0.74	-	-	-
	Min.	−6.1	−16.7	−4.5	−7.4
	Max.	8.7	5.1	7.2	0.3
ΔFwmax_r2-p [Nkg^−1^]	$\bar{x}$ ± s	1.3 ± 3.73	−2.5 ± 3.35	0.3 ± 2.04	−2.1 ± 2.78
	*Me ± Q*	-	-	-	-
	Min.	−6	−9.1	−3.2	−6.5
	Max.	8.5	3.62	4.6	4

**Table 3 ijerph-18-11680-t003:** Results of the ANOVA analysis of variance for repeated measures and the post-hoc test for changes in maximal relative strength compared to baseline in individual periods of the experiment in groups of participants.

Variable/Group	ΔFwmax_w-p [Nkg^−1^] vs. ΔFwmax_r1-p [Nkg^−1^]	ΔFwmax_w-p [Nkg^−1^] vs. ΔFwmax_r2-p [Nkg^−1^]	ΔFwmax_r1-p [Nkg^−1^] vs. ΔFwmax_r2-p [Nkg^−1^]
ISO-M	*p* < 0.001 ^N^ η^2^ = 0.82	*p* < 0.001 ^N^ η^2^ = 0.83	*p* = 0.82 ^N^ η^2^ = 0.18
ISO-P	*p* < 0.01 ^N^ η^2^ = 0.76	*p* < 0.05 ^N^ η^2^ = 0.76	*p* = 0.79 ^N^ η^2^ = 0.38
AUX-M	*p* < 0.001 ^P^ η^2^ = 0.85	*p* < 0.001 ^P^ η^2^ = 0.85	*p* = 0.98 ^P^ η^2^ = 0.05
AUX-P	*p* = 0.097 ^N^		

ΔFwmax_w-p—post-exercise change in relative strength following exercise relative to baseline, ΔFwmax_r1-p—change in relative strength after the first recovery period, relative to baseline, ΔFwmax_r2-p—change in relative strength after the second recovery period, relative to baseline, N—significance of differences tested by Dunn’s post hoc test, P—significance of differences tested by Tukey’s post hoc test. η^2^—measure of eta-squared effect size.

**Table 4 ijerph-18-11680-t004:** Results of the analysis of variance ANOVA (F-test; Tukey’s post-hoc test) and its counterpart for non-parametric ANOVA Kruskal–Willis tests (Dunn’s post hoc test) of the changes in maximum relative strength compared to baseline in subsequent periods of the experiment between groups of participants.

Variable/Group	ΔFwmax_w-p [Nkg^−1^]	ΔFwmax_r1-p [Nkg^−1^]	ΔFwmax_r2-p [Nkg^−1^]
ISO-M/ISO-P	*p* = 0.71 ^N^	*p* < 0.005 ^N^ η^2^ = 0.37	*p* < 0.001 ^P^ η^2^ = 0.39
ISO-M/AUX-M		*p* = 1.00 ^N^ η^2^ = 0.87	*p* = 0.73 ^P^ η^2^ = 0.08
ISO-M/AUX-P		*p* < 0.001 ^N^ η^2^ = 0.62	*p* < 0.005 ^P^ η^2^ = 0.40
ISO-P/AUX-M		*p* = 0.055 ^N^ η^2^ = 0.27	*p* < 0.05 ^P^ η^2^ = 0.37
ISO-P/AUX-P		*p* = 1.00 ^N^ η^2^ = 0.61	*p* = 0.98 ^P^ η^2^ = 0.48
AUX-M/AUX-P		*p* < 0.005 ^N^ η^2^ = 0.61	*p* = 0.05 ^P^ η^2^ = 0.38

ΔFwmax_w-p—post-exercise change in relative strength following exercise relative to baseline, ΔFwmax_r1-p—change in relative strength after the first recovery period, relative to baseline, ΔFwmax_r2-p—change in relative strength after the second recovery period, relative to baseline, N—significance of differences tested by Dunn’s post hoc test, P—significance of differences tested by Tukey’s post hoc test. η^2^—measure of eta-squared effect size.

## Data Availability

The data presented in this study are available on request from the authors. Some variables are restricted to preserve the anonymity of study participants.

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
