# Peer review of "Effect of Vibration Massage and Passive Rest on Recovery of Muscle Strength after Short-Term Exercise"

_ijerph, 2021, doi:10.3390/ijerph182111680_

Round 1

Reviewer 1 Report

General Comment

The authors have sought to compare the effects of low frequency vibration therapy on muscle function recovery. The manuscript is generally well written and easy to read and should be of interest to the journals readership. The introduction and discussion could be improved by considering whether the information and concepts presented can be written more concisely, for example the length of the discussion dedicated to the timing of vibration massage application which is beyond the scope of the data collected.

The authors have chosen to use the muscle contraction term auxotonic to refer to the cyclical performance of a single repetition. While this is not an incorrect term to use it is an uncommon term and it would improve the readability of the manuscript to a broader audience if a change in terminology was made, possibly just simplifying the reference to an isometric leg press repetition or a leg press repetition. The authors need to have a close read of the manuscript to correct formatting issues, such as there are two 'Table 1' and the use of Figures and Photos.

Specific Comment

Ln 13 and Table 1; The terms used to describe the participant groups and the abbreviations applied are inconsistent for the reader to gain an understanding of what has been conducted. It appears from the abstract that group abbreviations should be ISO,  AUX and CON or PAS better reflecting the exercise conditions as used. However when the reader understands Table 1 and the descriptive information, suggested group labeling should be ISO-M (Isometric plus massage), ISO-P (Isometric plus passive), AUX-M (Auxotonic plus massage), AUX-P (Auxotonic plus passive), 

Ln 118-119; The authors need to provide details as to when the maximal voluntary contraction (1RM) was performed after the warm up was completed so the 30% value should be stated that this was an estimated 30% value.

Ln 127; The authors need to include specific details on how the strength (force) in the MVC were measured. Was this with a load cell arrangement, or from a force plate, however regardless there is no details as to the sampling frequency, the type and style of measurement device and or the software used to record and analyse the signal. Alternatively details are required as to how the Rm strength assessments were conducted in the leg press.

Ln 136; The sentence needs to be re-ordered as the grammar is incorrect. Suggest rewording to, "consisting of 3 minutes isometric exercise following..."

Ln 176; Please change "Photo 1" to use 'Figure' and then re-number the rest of the manuscript as necessary.

Ln 274-275; Poor paragraph structure as a single sentence is not a paragraph and this statement either needs to be incorporated into another paragraph or deleted as it is not necessarily adding to the data interpretation and discussion.

Ln 284-296; Similar to the previous comment these lines include a series of single and two sentence paragraphs which is poor structure, reconsider these and combine or delete if they are not integral to the discussion and interpretation.

Ln 584 & 613; Check reference #61 and #77 for formatting

Author Response

Dear Reviewer,

Thank you very much for your time and valuable comments, which all have been considered and incorporated. The detailed list of responses is given below. We hope that the modifications and explanation will be acceptable for you.

Yours sincerely,

Rydzik, corresponding author

General Comment

The authors have sought to compare the effects of low frequency vibration therapy on muscle function recovery. The manuscript is generally well written and easy to read and should be of interest to the journals readership. The introduction and discussion could be improved by considering whether the information and concepts presented can be written more concisely, for example the length of the discussion dedicated to the timing of vibration massage application which is beyond the scope of the data collected.

The authors have chosen to use the muscle contraction term auxotonic to refer to the cyclical performance of a single repetition. While this is not an incorrect term to use it is an uncommon term and it would improve the readability of the manuscript to a broader audience if a change in terminology was made, possibly just simplifying the reference to an isometric leg press repetition or a leg press repetition. The authors need to have a close read of the manuscript to correct formatting issues, such as there are two 'Table 1' and the use of Figures and Photos.

A: Thank you corrected

Specific Comment

Ln 13 and Table 1; The terms used to describe the participant groups and the abbreviations applied are inconsistent for the reader to gain an understanding of what has been conducted. It appears from the abstract that group abbreviations should be ISO,  AUX and CON or PAS better reflecting the exercise conditions as used. However when the reader understands Table 1 and the descriptive information, suggested group labeling should be ISO-M (Isometric plus massage), ISO-P (Isometric plus passive), AUX-M (Auxotonic plus massage), AUX-P (Auxotonic plus passive), 

A: Thank you for your attention Corrected

Ln 118-119; The authors need to provide details as to when the maximal voluntary contraction (1RM) was performed after the warm up was completed so the 30% value should be stated that this was an estimated 30% value.

A: Prior to the study, a 1RM measurement of the athlete's maximal capacity was performed, and the 30% used in this study was matched to the previously determined maximal capacity of the athlete, so this is not an estimate but a value calculated from one maximal repetition with free weight

Ln 127; The authors need to include specific details on how the strength (force) in the MVC were measured. Was this with a load cell arrangement, or from a force plate, however regardless there is no details as to the sampling frequency, the type and style of measurement device and or the software used to record and analyse the signal. Alternatively details are required as to how the Rm strength assessments were conducted in the leg press.

A: The force measurement was performed according to the described procedure, selecting the weight by putting appropriate round plates on the measuring machine.

Ln 136; The sentence needs to be re-ordered as the grammar is incorrect. Suggest rewording to, "consisting of 3 minutes isometric exercise following..."

A: This has been corrected

Ln 176; Please change "Photo 1" to use 'Figure' and then re-number the rest of the manuscript as necessary.

A: This has been corrected

Ln 274-275; Poor paragraph structure as a single sentence is not a paragraph and this statement either needs to be incorporated into another paragraph or deleted as it is not necessarily adding to the data interpretation and discussion.

A: This has been corrected

Ln 284-296; Similar to the previous comment these lines include a series of single and two sentence paragraphs which is poor structure, reconsider these and combine or delete if they are not integral to the discussion and interpretation.

A: This has been corrected

Ln 584 & 613; Check reference #61 and #77 for formatting

A: This has been corrected

Reviewer 2 Report

Thank you for the opportunity to review the manuscript: Effect of vibration massage and passive rest on recovery of muscle strength after short-term exercise. This manuscript describes the effect of vibration massage and passive rest on the recovery of muscle strength after intense short-term exercise. The researchers showed that the vibration massage improves muscular recovery.

The paper is generally well written based on sound literature, the results well presented and discussed with respect to the literature.

The work is written following the steps of the scientific method.

The conclusions are the answer to the research question and result from the conducted research.

The study is well designed. However I have some minor comments I’d like to express.

Information about the participants, was their level of physical activity determined, were they active athletes? what were the specific criteria for inclusion in the study?

P-3, lines 105 - 109 - please correct the punctuation marks

P-5 - In my opinion, Photo 1, can be removed

line 177 - incorrect paragraph numbering, it should be 2.4

The article is generally valuable and correctly written, please treat the above comments only as suggestions.

Author Response

Dear Reviewer,

Thank you very much for your time and valuable comments, which all have been considered and incorporated. The detailed list of responses is given below. We hope that the modifications and explanation will be acceptable for you.

Yours sincerely,

Rydzik, corresponding author

Thank you for the opportunity to review the manuscript: Effect of vibration massage and passive rest on recovery of muscle strength after short-term exercise. This manuscript describes the effect of vibration massage and passive rest on the recovery of muscle strength after intense short-term exercise. The researchers showed that the vibration massage improves muscular recovery.

The paper is generally well written based on sound literature, the results well presented and discussed with respect to the literature.

The work is written following the steps of the scientific method.

The conclusions are the answer to the research question and result from the conducted research.

The study is well designed. However I have some minor comments I’d like to express.

Information about the participants, was their level of physical activity determined, were they active athletes? what were the specific criteria for inclusion in the study?

A: The inclusion criteria for the study were consent and willingness to participate and a positive medical examination. The subjects were not active athletes; however, they practiced recreational physical activity with similar levels of intensity and volume determined prior to the study. Additionally, the criteria presented are described at the beginning of the material and method section.

P-3, lines 105 - 109 - please correct the punctuation marks

A: This has been corrected

P-5 - In my opinion, Photo 1, can be removed

A: Photo 1 depicts the nature of the study so we propose to leave it

line 177 - incorrect paragraph numbering, it should be 2.4

A: This has been corrected

The article is generally valuable and correctly written, please treat the above comments only as suggestions.

Round 2

Reviewer 1 Report

The authors have made some changes as requested but further edits are required to the Methods section to provide the reader with a clear description of the procedures so that the investigation could be replicated if desired.

Ln 115-119 and Figure 1; No reference is made to when the 1RM assessment is performed although the authors indicated in their reply to the first review that this was conducted prior.

Ln 127-132; The authors indicate that maximal isometric force is assessed but there are no details of what device is used only a reference is made to the leg press device. The results are report in Nkg but if I understand correctly the measured unit was kg of mass lifted but this assumption is not practical in an isometric contraction. Thus a clearer description of the assessment procedure is required.

Ln 262; The 1 needs to be superscript

Author Response

Dear Reviewer,

Thank you very much for your time and valuable comments, which all have been considered and incorporated. The detailed list of responses is given below. We hope that the modifications and explanation will be acceptable for you.

Yours sincerely,

Rydzik, corresponding author

The authors have made some changes as requested but further edits are required to the Methods section to provide the reader with a clear description of the procedures so that the investigation could be replicated if desired.

Ln 115-119 and Figure 1; No reference is made to when the 1RM assessment is performed although the authors indicated in their reply to the first review that this was conducted prior.

A: This has been corrected

Ln 127-132; The authors indicate that maximal isometric force is assessed but there are no details of what device is used only a reference is made to the leg press device. The results are report in Nkg but if I understand correctly the measured unit was kg of mass lifted but this assumption is not practical in an isometric contraction. Thus a clearer description of the assessment procedure is required.

A: This has been corrected

Ln 262; The 1 needs to be superscript

A: This has been corrected